# Position:
# Agentic Systems Constitute a Key Component of Next-Generation Intelligent Image Processing

## Abstract

This position paper argues that the image processing community should broaden its focus from purely model-centric development to include agentic system design as an essential complementary paradigm. While deep learning has significantly advanced capabilities for specific image processing tasks, current approaches face critical limitations in generalization, adaptability, and real-world problem-solving flexibility. We propose that developing intelligent agentic systems, capable of dynamically selecting, combining, and optimizing existing image processing tools, represents the next evolutionary step for the field. Such systems would emulate human experts' ability to strategically orchestrate different tools to solve complex problems, overcoming the brittleness of monolithic models. The paper analyzes key limitations of model-centric paradigms, establishes design principles for agentic image processing systems, and outlines different capability levels for such agents.

## 1 Introduction

Image processing is a longstanding research area in computer vision. We have a wide variety of image processing and editing needs, ranging from post-photography editing, image restoration, enhancement, to style transfer. These tasks are inherently complex due to both the intricate nature of images and the unique aesthetic standards and nuanced expectations that humans hold. For a long time, image processing has been a specialized technical field managed by

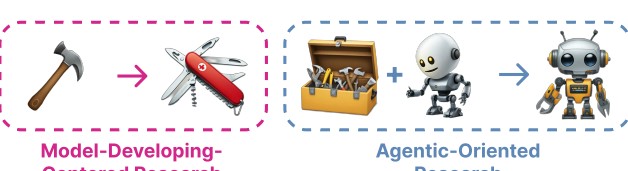

Figure 1: The existing research paradigm focuses on developing more powerful and multi-functional image processing models. In contrast, we advocate a new research paradigm centered on building agentic systems. Our goal is to create an agent that can integrate and leverage these models to achieve higher levels of intelligence, automation, and generality.

dedicated technicians and artists. Efforts in computer vision have long aimed to provide high-quality tools that enhance the efficiency and effectiveness of image processing tasks. *The research community strives to develop intelligent, adaptable software that maximizes convenience for users at all levels and fulfills a wide range of image processing needs.*

Early image processing algorithms were typically designed for specific types of problems, making them part of a broader pipeline or a standalone tool [5]. Professionals often need to configure and combine multiple processing steps to address particular image processing challenges. In the past decade, deep learning has driven a major leap in image processing, significantly improving the quality of individual tasks while introducing a more generalized, intelligent paradigm [14, 65]. The

industry has gradually shifted from constructing image processing pipelines to training end-to-end deep learning models that replace complex pipelines [3]. These deep learning models are now used not only to solve isolated problems but also to establish a general, multi-task, and intelligent solution that can operate effectively in diverse, real-world conditions [31, 50, 62, 68]. The advent of deep learning and artificial intelligence has made the vision of a general, intelligent, software-based "image processing assistant" seem closer than ever, though it remains just out of reach. **Significant research efforts have focused on the current paradigm, which is predominantly centered on developing various deep image processing models, as shown in Figure 1 (left).**

Nevertheless, the limitations of deep networks are emerging, and continuing within the existing research paradigm makes overcoming these constraints challenging. Firstly, these models face issues with generalization, as they perform well on test data similar to their training data but struggle on test data that deviates significantly from it [20]. Secondly, deep models capable of handling a wide range of degradation scenarios often compromise on quality and generalization [66]. Those that excel in specific degradations may lack generalizability, while models that handle a broad spectrum of tasks may not deliver peak performance on any single task [31, 67]. These challenges suggest that relying on a single model or fixed process for image processing may not effectively address the dynamic and complex real-world problems. Interestingly, despite the limitations of current image processing models, human artists and image editing professionals can still leverage these models and tools – often very simple ones, like basic operations in PhotoShop – to accomplish complex tasks that even the most advanced models cannot achieve. Emulating the dynamic and adaptive ways in which humans use these image processing tools could be a crucial step toward making image processing more intelligent and general. After all, no matter how powerful or multi-functional a tool may be, it still requires a capable operator – human or otherwise – to realize its true potential, see Figure 1 right.

**In this position paper, we advocate for a new research paradigm centered on agentic-oriented image processing systems, offering a more autonomous, adaptable, and intelligent alternative to current methods.** We begin by discussing the core capabilities required for intelligent image processing systems and the challenges that the current paradigm faces in Sec. 2. We then introduce the concept of AI Agents in Sec. 3, exploring their fundamental principles and role in intelligent systems. In Sec. 4, we extend this discussion to agentic image processing systems, analyzing how existing methods can incorporate varying degrees of agentic features to enhance generality and intelligence. We also examine the key characteristics of agentic image processing systems and outline different levels of agentic capability. Recognizing that large language models have become pivotal in the study of intelligent agents, we also explore how language models and multi-modal techniques may shape the future of image processing. Moreover, in Sec. 5, we highlight that there remains further room for exploration in certain critical attributes that determine a system's level of intelligence and generalization.

**Alternative Views.** The prevailing view holds that continued progress in developing models – through scale and improved architectures – could eventually overcome all the limitations and subsume the proposed agentic capabilities. While these views have merit, they underestimate the fundamental mismatch between static model architectures and the dynamic, compositional nature of real-world image processing requirements. Hybrid approaches combining foundation models with agentic components may offer a viable middle ground, but system intelligence requires explicit architectural support beyond current paradigms.

## 2 Backgrounds

### 2.1 Intelligent Image Processing

The core of intelligent image processing lies in developing intelligent and efficient algorithms that enable computers to automatically and accurately process images in various conditions to meet visual, psychological, or other needs. Ultimately, its goal is to build a "software employee" capable of automatically, intelligently, and effectively completing various image processing tasks. This is a highly visionary goal, and the community has long been approaching it from different angles in an attempt to simplify this challenging problem. Initially, image processing methods were mainly extensions of signal processing techniques applied to two-dimensional image signals, focusing on specific image processing operations. Entering the 21st century, with advancements in computing ability, many methods based on image priors and optimization have emerged but remain limited to

specific task scenarios. In the past years, the rise of machine learning and deep learning [32, 14] has propelled significant progress in the field of image processing. Particularly, the introduction of neural networks has led to breakthrough results in various image processing tasks. Data-driven methods not only allow for multiple image tasks to be handled within the same algorithmic framework but also make multitask integration and general-purpose image processing possible. Researchers have once again embraced the vision of intelligent image processing, and constructing a "software employee" capable of handling all tasks by unifying image processing tasks seems to have become a feasible direction. This position paper focuses on the core challenges of realizing this vision and the approaches to overcome them.

Specifically, an intelligent image processing system, a "software employee", needs to possess at least the following core capabilities:

- *Generality*: The system should be able to handle a wide range of diverse tasks without requiring separate models for each one, nor relying on extensive domain-specific training data or explicit task-specific instructions.

- *Autonomous*: The system should minimize reliance on user operations and supervision. It demonstrates proactivity by leveraging prior experience to explore new strategies without requiring explicit instructions.

- *Intelligence*: The system can adaptively adjust its processing strategies based on the semantic content and quality of the input, and user instructions. The system should demonstrate its intelligence and complexity in the processing workflow or in the final outcomes.

- *User Interaction and Feedback*: The system should facilitate clear, continuous, and user-friendly communication.

- *Self-Evolution and Creativity*: The system should generate meaningful and innovative solutions, going beyond straightforward problem-solving to provide novel approaches, insights, or outputs that showcase originality. Additionally, the system can continuously evolve and learn from new data, experiences, and user interactions.

## 2.2 Why Intelligent Image Processing is Challenging?

However, the current mainstream research paradigm, which centers around the development of deep learning models, struggles to align with the aforementioned vision.

**The Challenge of Achieving Generality.** Unlike high-level image understanding tasks, image processing tasks have both input and output as images that require precise pixel-level correspondence. The information needed for image processing tasks is not as specific as in image understanding. In image understanding, the model abstracts the image, extracts main features, and aligns them with semantics expressible in human terms. Although we hope that advanced deep networks for image processing can also learn the "semantics" of images, it is challenging to accurately describe local image details semantically. In fact, image processing networks do not learn semantics [19, 37, 22] but instead learn certain image transformations and overfit to training degradations [20, 39]. This is determined by the training paradigm of deep image processing models. Therefore, essentially, current image processing networks are not intelligent.

This leads to the next issue: the differences between various image processing tasks are also distinct from other types of tasks. Generally speaking, a specific image transformation or degradation can define an entirely new task. The differences between image transformations or degradations can be very subtle, and they can also be compounded to create almost unlimited types of transformations or degradations, resulting in virtually infinite image processing tasks. Due to deep models overfitting to the training set [8], tasks beyond the training scope cannot be well addressed [38]. This greatly limits the ability of current image processing methods to solve general problems. Worse yet, because collecting training images in the real world is extremely difficult, most research can only train on synthetic data, which further leads to generalization issues in practical applications.

Some methods attempt to include as many tasks as possible in the training set and train a sufficiently large model to achieve generality for common tasks [31, 11], even hoping that increasing the number of tasks will enable the network to generalize. However, these models have been proven to have a trade-off between the range of tasks and processing performance [66]. It's challenging to expand

the task range while keeping image processing performance from significantly declining. All these issues make constructing image processing systems with general capabilities highly challenging.

**The Challenge of Developing Intelligence.** Beyond the requirements of generality, we are increasingly emphasizing the intelligence these image processing systems exhibit. Firstly, we hope that image processing systems can explicitly perceive image content and perform targeted processing based on that content. For example, generating corresponding fur on animals or inferring and completing blurry or missing objects. Existing research indicates that end-to-end supervised deep image processing models do not possess this characteristic [37], but methods based on pre-trained generative models have demonstrated related capabilities and have thus achieved good results [62]. Secondly, we expect intelligent image processing systems to adaptively adjust processing strategies based on different input types or qualities, and even have the ability to make complex decisions based on specific image content. For instance, the system can automatically select the optimal denoising, enhancement, or restoration methods according to the image's resolution, lighting conditions, or noise levels. Additionally, we hope that image processing systems can dynamically understand users' complex needs. Currently, users need to select tools and set parameters based on their own expertise before obtaining results; this process does not reflect the system's intelligence. An intelligent system can accept user feedback or instructions to make dynamic adjustments in subsequent processing. These requirements have been mentioned to varying degrees in image processing research, but none have been explored in depth.

**The Challenge of Balancing Autonomy, User Interaction, and Creativity.** Existing approaches often fall into two extremes. On one hand, fully autonomous methods – such as end-to-end models – can quickly complete tasks but tend to overlook subtle user preferences, resulting in a rigid, one-size-fits-all automation. Automatic denoising may eliminate intentionally added artistic grain, and style transfer algorithms can homogenize diverse creative visions. Given the broad range and complex demands of image processing tasks, achieving consistently high-quality results proves challenging with these models. The end result is that people still need to pick and combine the results of different models, thus losing this automaticity. On the other hand, heavily manual interfaces impose a significant technical burden on users. Professional software like Photoshop requires extensive manual intervention and expert knowledge, which conflicts with the goals of ease of use and accessibility. Moreover, many existing approaches rely on single-model solutions with limited interactivity; more semantic, higher-level, and varied interaction mechanisms are needed to facilitate seamless communication between the user and the system.

Furthermore, existing methods also struggle to foster genuine creativity. Here, "creativity" goes beyond generating novel content via generative models [62]; it also involves discovering innovative ways to repurpose existing tools and deepening our understanding of them. As image processing evolves from mere technical correction into a creative medium, bridging this gap demands systems that not only "see" the pixels but also interpret the cultural, emotional, and contextual layers – a frontier that remains largely unexplored in current technology.

## 3  What is AI Agent?

An agent is a program designed to achieve its goals by perceiving the environment and interacting with it through available tools. These agents can operate autonomously without human intervention and proactively work towards their objectives [17, 34, 56]. From a design perspective, agent-based systems naturally fulfill our demand for automation. The various image processing models we develop can be regarded as tools of different scales and purposes, while the agent acts as the "coordinator" that actively orchestrates these tools, as shown in Figure 1 left. Early agent programs largely relied on symbolic methods [17] and reinforcement learning [24, 46, 63]. In recent years, however, agent systems powered by Large Language Models (LLMs) have achieved transformative progress [58, 7]. By training on massive text corpora through next-token prediction, LLMs demonstrate powerful knowledge transfer and logical reasoning abilities [4, 41, 1, 47, 25, 15], showcasing considerable potential in complex reasoning [51, 30], step-by-step planning [57, 58], and domain-specific knowledge applications [40, 21]. Compared to traditional reinforcement learning agents, LLM-based agents maintain long-term planning and simultaneously leverage broad general knowledge, thereby exhibiting more human-like cognitive characteristics [1]. From a cognitive standpoint, the fundamental responses of an LLM can be likened to "System 1," characterized by rapid, automatic thinking, whereas more advanced composite agent systems emulate "System 2,"

which involves deliberate, reflective reasoning [57, 35, 28]. Recent research has explored diverse agent architectures that enhance LLM-based problem-solving through structured mechanisms [43, 49] – such as tree- or graph-based search strategies [2, 57], external tool integration [44, 52], memory retrieval systems [70, 42], and error-driven learning processes [45, 58]. By combining an LLM's reasoning capabilities with structured problem-solving frameworks, these approaches show strong potential for tackling complex tasks [16].

Notably, pioneering efforts have employed LLM agents across various domains, all striving to create automated systems capable of proactively tackling a broad spectrum of challenges, aligning with the vision outlined in this position paper. For instance, frameworks like HuggingGPT [44] and Visual ChatGPT [44] leverage LLMs as multi-modal task controllers, integrating them with model libraries to decompose and solve diverse tasks; frameworks like OctoPack integrate LLMs with specialized toolsets, achieving significant performance gains in fields like medical image processing [40]. Advancements have also highlighted the effectiveness of LLM agents in tackling complex image processing tasks, achieving remarkable results [69, 9]. These advancements collectively highlight the transformative potential of LLM-based agents in addressing complex multi-modal challenges.

# 4   Agentic Image Processing System

The initial step toward agentic image processing involves acknowledging the fundamental reality that, **regardless of how advanced your image processing model is, carefully chosen preprocessing, postprocessing, or application-specific techniques/tricks can often enhance its performance**. For instance, certain severe degradations cannot be fully restored by a single pass through an image restoration model; applying the model iteratively to its own outputs can yield further improvements. Additionally, some degradations may lie beyond the training scope of the model, and introducing deliberate additional blurring before restoration can significantly mitigate these challenging cases. There exist numerous possibilities for such operations, and in practical applications, users frequently leverage these techniques to maximize performance.

## 4.1   Paradigms of Current Image Processing System

While this paper is the first to advocate for the construction of an agentic system to address challenges in intelligent image processing, traces of agentic thinking have already emerged, to varying degrees, in previous studies. We begin by examining the embodiment of agentic concepts behind the design of existing methods, adopting a perspective that progresses from simple to complex. Figure 3 provides a schematic illustration of these paradigms, offering a visual aid for better understanding.

**End-to-End** models are the most common paradigm in image processing research. Given an input, an end-to-end model produces a corresponding output. This category encompasses optimization-based, filter-based, and deep network models, with a focus on end-to-end deep network models for intelligent models. The standard approach involves collecting images that need processing along with their corresponding target images to form training image pairs, and then training the image processing model on this basis. This paradigm is the least agentic, and due to the following reasons, it has limitations in terms of generality and intelligence: Due to the limitations discussed in Sec. 2, no single model can simultaneously achieve both broad image processing capabilities and outstanding results. If a model is designed to be sufficiently "general," it will inevitably come at the cost of reduced performance on specific tasks.

**Pipeline** paradigm typically decomposes complex and difficult-to-model-at-once image processing problems into multiple independent processing steps. The main advantage of this approach is that it can effectively break down complex tasks into more manageable subtasks, allowing for the creation of new tasks through the combination of a limited number of image processing/operations [5, 3]. The modular design also equips the pipeline paradigm with high flexibility and scalability, enabling the system to be adjusted and updated according to specific needs. This makes it convenient to integrate new technologies or algorithms into the existing framework. For example, users can directly replace the denoising step with the latest denoising algorithm without redesigning the entire pipeline. Pipeline design is a typical idea of people to solve complex problems by combining simple tools.

Although pipeline models have advantages in handling complex tasks, their agentic level is still relatively low because each step is pre-defined based on practical applications and is difficult to adjust

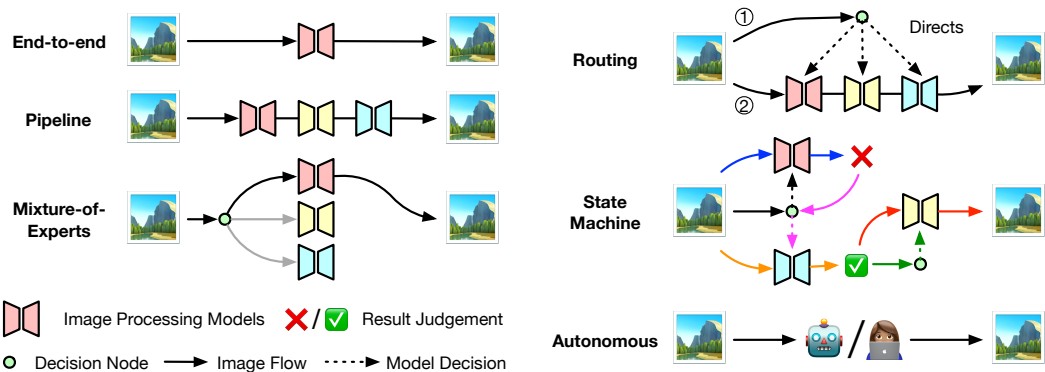

Figure 2: How image processing systems can embody different levels of agentic to enhance their generality and intelligence.

according to the diversity of inputs. Moreover, pipeline models often rely heavily on manual design, as the task decomposition and execution order within the pipeline significantly impact the results [63, 9, 69]. Therefore, they are used for specialized solutions to specific real-world problems rather than aiming for pursuing generality or higher levels of intelligence. However, by integrating different steps, the pipeline approach expands the application boundaries of image processing algorithms. This characteristic is a core advantage that agentic image processing systems can leverage.

**Mixture-of-Experts (MoE)** is another paradigm that broadens the task range of image processing systems and enhances performance by integrating the capabilities of multiple models. A single model is constrained by the trade-off between task coverage and processing effectiveness, making it difficult to efficiently handle each individual task while covering a large number of tasks. Similar phenomena have been observed in other large-scale model practices [6, 13, 12, 26]. To overcome these limitations, MoE typically introduces multiple expert models, each focusing on a specific task. The system dynamically selects the most suitable expert based on the input data or the task requirements, or combines the outputs of multiple expert models to optimize processing performance. This approach not only achieves a balance between task coverage and processing effectiveness but also allows for flexible adaptation to new tasks or improvement of performance on specific tasks by adjusting and replacing expert models. Therefore, MoE becomes an effective means to achieve task breadth while ensuring processing depth. Although MoE achieves a certain degree of agentic through proactive model selection, its generality and intelligence still depend on the performance of each expert model within the system. Since we cannot infinitely expand the number of expert systems, and there still exist problems that individual models cannot effectively solve, the generality of the MoE paradigm remains quite limited.

**Routing** is a combination of the Pipeline and MoE paradigms, potentially integrating the advantages of both. Similar to the MoE paradigm, the routing paradigm selects corresponding processing paths for input images to achieve targeted processing. However, unlike MoE, the routing paradigm selects a Pipeline composed of multiple models to maximally expand the range of feasible tasks. In essence, the routing paradigm automatically devises dedicated pipelines for different input image tasks and invokes the corresponding models. In other words, routing makes a "plan" for each input and executes it [23, 63]. The routing paradigm further enhances the system's agentic; when the decision-making methods are sufficiently accurate and robust, this paradigm can greatly expand the potential task coverage, thereby improving its generality. Since the decision-making process requires a deeper understanding of the images, the routing paradigm also possesses higher intelligence. However, once the path is determined, the outcome is already fixed. If an issue arises in an intermediate step, the routing approach cannot backtrack to address the problem at that specific step.

**State Machine.** Building on the foundation of the routing paradigm, state machines further expand to allow more fine-grained control over the processing flow. Similar to routing, a state machine produces a complex execution plan to conduct multiple image processing steps. However, due to the complexity of images and the variety of image processing operations, it is often not feasible to directly determine an optimal plan or parameter set in one run. In contrast to routing, the most notable feature of a state machine is its intelligent flow control: the system can reason and autonomously decide whether to proceed to the next step, adopt the current result or plan, or even undo the previous operation.

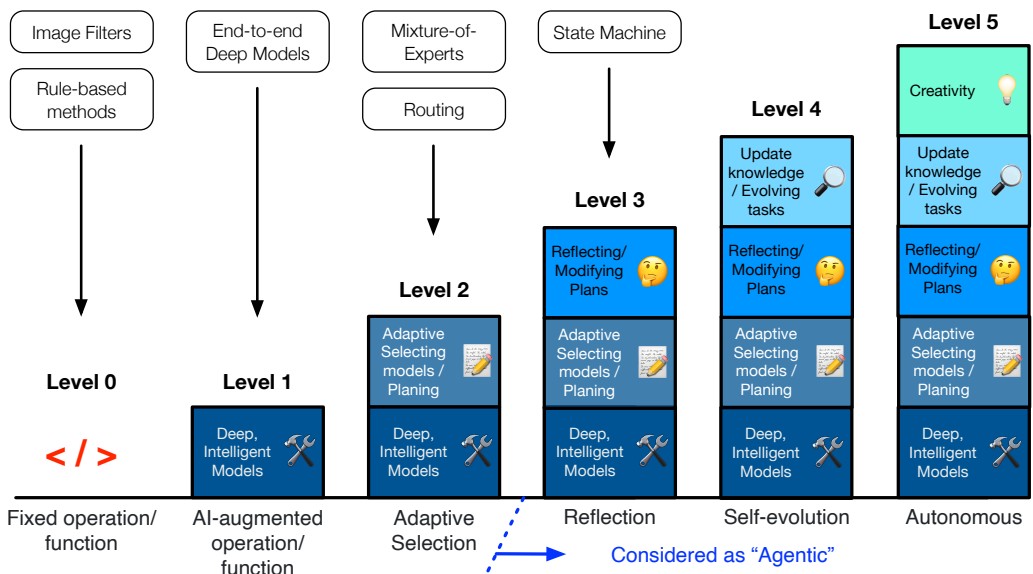

Figure 3: Levels of agentic capability in image processing systems, illustrating the progression from fixed rule-based methods (Level 0) to fully autonomous and creative systems (Level 5). Each level builds on the previous one, adding layers of adaptability, reflection, self-evolution, and creativity.

Essentially, the process by which humans solve problems can also be viewed as a highly flexible state machine. Some pioneering studies have already adopted this paradigm to build intelligent agent systems, demonstrating their remarkable intelligence and potential [69, 9, 36].

## 4.2 Characteristics of an Agentic System

Based on the above analysis, several core design principles of agentic thinking are already present in prior works to varying degrees. To further advance this approach, we summarize below the potential features of an agentic system – features that can significantly improve the system's intelligence, generality, and ease of use:

- *Proactive and Autonomous Problem-Solving*: An agentic system autonomously senses challenges, explores different models and methods, and dynamically adjusts strategies without relying on further human instructions. This allows for flexible and efficient image processing even in complex scenarios.

- *Integration of Multiple Models/Tools*: Rather than depending on a single model for complex image processing tasks, an agentic system can strategically combine multiple models or tools according to the specific task requirements or image characteristics. Even "all-in-one" large models can be combined with other operations or models to broaden coverage and improve performance.

- *Adaptive, Context-Aware Strategies*: An agentic system tailors its processing strategy based on the specific content or characteristics of the input image instead of applying a fixed pipeline. In other words, the system reasons about the image to make informed decisions.

- *Modular Architecture*: An agentic system often consist of multiple functional modules that work together – commonly including Perception, Reasoning, Action, and Reflection. Data and results flow through these modules, forming a coherent and synergistic workflow.

- *Easy Extensibility*: An agentic system should be readily extensible, allowing new features, tools or modules to be added without large-scale retraining or constructing massive new datasets. This flexibility enables the system to adapt to evolving requirements more effectively.

- *Continuous Reflection and Improvement*: An agentic system incorporates reflection mechanisms to evaluate and refine their performance. This iterative learning process ensures the system can leverage real-world usage data for sustained improvement over time.

## 4.3 Levels of Agentic in Image Processing

It is clear that "agentic" is not a binary concept but rather exists on a continuum. Drawing inspiration from the levels of autonomous driving [29], we propose a reference framework for classifying the agentic levels of image processing systems into six tiers, as shown in Figure 2. These levels reflect different characteristics that such systems may exhibit:

- **Level 0 (Fixed operation/function).** Methods at this level only provide basic, fixed transformations and processes. Regardless of the input image, they perform operations strictly according to predefined rules, such as filter-based or rule-based transformations. This stage exhibits almost no intelligence or generality.
- **Level 1 (AI-augmented operation/function).** While still focused on specific tasks, systems at this level go beyond simple rules by incorporating complex patterns learned through deep learning or other data-driven approaches. Although their performance surpasses Level 0, they remain limited in generalization.
- **Level 2 (Adaptive Selection).** Starting from this level, image processing systems no longer rely on a single model or tool. Instead, they can adaptively select and integrate different models, thus expanding the range of tasks they can handle. The ability to choose different processing strategies based on the input image demonstrates a certain degree of intelligence.
- **Level 3 (Reflection).** This level introduces more freedom in workflow control and the ability to reflect on output results, allowing systems to flexibly adjust strategies and processes. Through reflection and iterative adjustments, systems at this level can tackle a wider variety of problems.
- **Level 4 (Self-evolution).** At this level, agentic surpasses what fixed architectures can achieve. Systems can continuously learn and evolve from large amounts of data and experience, distilling new knowledge to solve previously unsolvable problems. They may even modify or advance their own workflows.
- **Level 5 (Fully Autonomous).** At the highest level, agents can execute image processing tasks autonomously without user intervention. In addition to incorporating all capabilities of the previous levels, they possess a degree of creativity, enabling innovative problem-solving approaches (e.g. discovering new tricks that the people who created the agent don't know about.). As a result, they can potentially replace human experts and approach a form of artificial general intelligence (AGI).

## 4.4 The Role of LLMs

Incorporating agentic design enables greater autonomy and adaptability. Current image processing paradigms can partially fulfill these objectives if designed with sufficient complexity (e.g., approaches based on reinforcement learning [23] or expert systems [64]). However, due to the inherent complexity of image processing tasks and the ambiguity of their descriptions, existing paradigms struggle to further develop and leverage agentic capabilities. LLMs have demonstrated strong adaptability when dealing with open-domain problems.

When an agent autonomously tackles complex tasks, it often faces numerous scenarios arising from the interplay of diverse factors. Given the complexity of image processing systems, these scenarios cannot be easily summarized or handled with simple rules. Traditional methods rely on predefined rules and features, which become insufficient in the face of combinatorial explosions. In contrast, LLMs can perform language-based reasoning and planning for each situation, offering remarkable generalization capabilities. Through this reasoning mechanism, abstract and unstructured demands can be mapped to specific image processing models or tools and translated into executable steps. Moreover, the workflow can be dynamically adjusted based on real-time feedback – for instance, rolling back or modifying the previous step. LLMs may even derive innovative new model combinations.

In a multi-modal setting, LLMs further provide the system with an "intelligent eye," enabling it to extract semantic information at multiple levels from abstract visual signals, far beyond what non-LLM approaches can achieve [61, 60, 53, 59]. Finally, natural language dialogue has proven to be an efficient and user-friendly channel for interaction. By employing LLMs, the system can engage in more flexible conversations with users, offer feedback, and accept instructions, thereby significantly enhancing both usability and extensibility.

## 5 Core Problems Demanding Further Study

Building intelligent agentic systems is a novel direction with many core challenges that require in-depth exploration. We analyze key issues that warrant attention in future research. Due to space constraints, we focus on two potential directions here and discuss additional topics in the appendix.

### 5.1 Cognitive Architecture of Image Processing

The foundation for building more complex agentic systems lies in designing their overall **cognitive architecture**[1]. A cognitive architecture refers to how the system "thinks" – in other words, the flow of code, prompts, and LLM calls that accept user input and execute operations or generate responses. Designing a cognitive architecture involves contemplating the abstract processes by which an intelligent agent solves problems at different levels. It's the methodology an agent uses to address a certain class of problems.

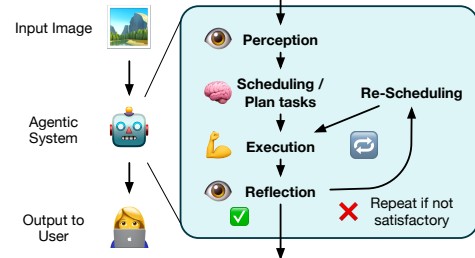

Figure 4: Cognitive architecture for image processing systems, illustrating the iterative process of perception, scheduling, execution, reflection, and rescheduling to achieve satisfactory results.

To facilitate understanding, we can start with the abstracted process of humans performing image processing or PhotoShop editing tasks. Figure 4 illustrates an example of a personified cognitive architecture for an image processing system, which is also the architecture used by Zhu et al. [69] and Chen et al. [9]. In this architecture, the interaction between the system and tools is abstracted into five stages: **Perception, Scheduling, Execution, Reflection, and Rescheduling**. Specifically, the Perception stage acts as the agent's "eyes," extracting necessary information from the input image. The Scheduling stage functions like the "brain," making judgments and formulating plans based on the acquired information and existing knowledge. The Execution stage represents "action," carrying out specific operations according to the plan. The Reflection stage evaluates whether the intermediate results meet expectations. If they do, the agent proceeds with subsequent plans; if not, the Rescheduling stage considers the failed results and modifies the original plan.

However, this intuitive cognitive architecture still leaves much room for improvement in the architectural research of image processing agent systems. For instance, for more specific problems, how should we design their cognitive architectures to meet the need for more refined control? For tasks requiring higher generality, how can we abstract a sufficiently general process to encompass a wider range of possible tasks? How can we systematically explore, distill, and abstract human problem-solving strategies into foundational principles for designing cognitive architectures? Furthermore, how can we create a cognitive architecture that surpasses human limitations, optimized specifically for intelligent image processing agent systems?

### 5.2 More Problems Demanding Further Study

Due to space limitations, we discuss four additional topics in the appendix: *Image Quality Assessment and Content Analysis*, *Knowledge Acquisition and Infusion*, *Human-Computer Interaction in Agentic Systems*, and *Exploitative Learning, Self-Evolution & Creativity*.

## 6 Conclusion

The evolution from task-specific models to agentic image processing systems marks a fundamental shift in addressing real-world complexity through dynamic tool orchestration rather than monolithic architectures. By embedding human-like adaptive reasoning into operational frameworks, such systems transcend current generalization limits while preserving specialized model strengths.

---

[1]The term "cognitive architecture" has a rich history in neuroscience and computational cognitive science. It refers both to theories about the structure of human thought and to computational implementations of these theories. Here, we borrow this concept but do not specifically refer to its original meaning.

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

# Appendix

# A  More Problems Demanding Further Study

## A.1  Image Quality Assessment and Content Analysis

Regardless of how we design our cognitive architecture, certain fundamental capabilities are indispensable. Among these, the recognition, analysis, and evaluation of image content and quality are essential. In Figure 4, the Perception and Reflection stages are related to this capability; they

serve as the "eyes" of the image processing agentic system. Historically, image content recognition [48] and image quality assessment [27, 18] have been independent research fields, separate from image processing models, each with its own methods and objectives. However, from the research perspective of agentic systems, we impose higher demands on them.

For image content recognition, we need to consider its robustness under different image qualities and special circumstances. The recognized content must be designed with finer granularity to meet the specific needs of image processing, rather than being overly abstract like high-level vision tasks. Regarding image quality assessment, we cannot limit ourselves to evaluating a single "score." Models need to be more intelligent, performing fine-grained image quality analysis, determining types of distortion, and assessing the quality of intermediate results. The decisions of the entire agentic system largely depend on the accuracy and intelligence of this pair of "eyes."

Thanks to the development of multi-modal language models, a number of tools have now begun to demonstrate this capability [61, 60, 53, 54, 55]. Utilizing language models, these methods can describe image content, analyze image quality, evaluate the pros and cons of different image qualities, and provide judgments based on quality. These methods have already been applied in early research on agent-based image processing systems, showcasing their potential. However, the intelligence and accuracy of these methods still have a significant gap compared to large-scale practical applications.

## A.2 Knowledge Acquisition and Infusion

After discussing the "eyes," let's turn to the "brain." As previously analyzed, the planning and decision-making abilities in intelligent agentic systems mainly stem from language models, which base their decisions on general knowledge learned from large volumes of text training data. Only when a language model has encountered problems and knowledge related to image processing during training can it be expected to make accurate judgments in the system; otherwise, the language model may struggle to provide reliable predictions. However, pre-trained language models usually contain only the most basic knowledge. If we want an image processing system to perform more specific and precise tasks, we need to supply the language model with the necessary knowledge. This involves two issues: the acquisition of knowledge and the injection of knowledge.

Firstly, acquiring such knowledge is non-trivial, and the method of injecting this knowledge into the system depends heavily on how it is represented. Image processing involves not only a large amount of conceptual and systematic knowledge but also a wealth of experience-based and case-based knowledge. This kind of knowledge is difficult to abstract into rules and usually exists in the form of case studies. Early attempts mainly employed two methods. Chen et al. [9] collected a series of input images along with corresponding instruction-output training data to implicitly carry a large amount of knowledge and information. The trained model then has the ability to handle similar problems. However, the drawbacks of this method are evident: firstly, collecting a large amount of high-quality training data requires substantial resources and is both costly and difficult. Secondly, the model lacks scalability; adding new knowledge requires retraining the model. Additionally, fine-tuning the language model may compromise its general capabilities.

In contrast, Zhu et al. [69] rely on the reasoning ability of an unmodified language model and provide a reference "manual." This method of supplying knowledge is known as Retrieval-Augmented Generation [33]. They first use the language model to summarize a large amount of scattered case information into knowledge that can be described linguistically. When solving actual problems, they provide the relevant content together. The language model utilizes its zero-shot learning and contextual inference capabilities to complete tasks based on the provided information. The advantage of this method is that it does not fine-tune the model, avoiding the loss of general performance, and the model still possesses strong reasoning and understanding abilities. However, its drawbacks lie in the difficulty of accurately describing professional knowledge in language. Moreover, quickly retrieving relevant information from a vast amount of knowledge is not easy, and it also places high demands on the design of the cognitive architecture.

In this area, we currently have only very preliminary results. The exploration, acquisition, representation, and injection of prior knowledge in image processing will become the core research topics of image processing agent systems.

### A.3 Human-Computer Interaction in Agentic System

Agentic systems' multi-step operational paradigms, randomness, and natural language interfaces introduce new challenges in human-computer interaction. Currently, the primary way to interact with intelligent agent systems is through "chatting," where the system communicates its thoughts and actions in a conversational manner. However, we need new interaction methods to meet higher demands.

We must provide users with visibility into what the agent is doing by displaying all the steps it takes, allowing users to observe and understand the ongoing processes. Simultaneously, users should be able to give the agent more fine-grained and explicit instructions to control its behavior more precisely. Moreover, users should not only see what is happening but also have the ability to correct the agent. If they discover that the agent made an incorrect choice at step four (out of ten), they should be able to return to that step, correct the agent in some manner, and then proceed with the execution. The ultimate goal is to achieve collaboration between the agent and the user, enabling them to complete tasks together effectively.

### A.4 Exploitative Learning, Self-Evolution & Creativity

The development of agentic systems has opened up new horizons in the fields of exploitative learning, self-evolution, and creativity. These concepts are crucial for advancing intelligent systems, enabling them to autonomously adapt, improve, and innovate over time without explicit human intervention, achieving higher levels of automation and agency as depicted in Figure 3.

Exploitative learning refers to the agent itself taking the initiative to determine the methods and content of knowledge acquisition within certain limits. The work of Chen et al. [10] embodies the prototype of this idea: they presented many experimental results to the agent, and the agent selected valuable content from them to learn. In some cases, the agent could even take some unconventional actions to acquire new knowledge through interaction with the world.

Self-evolution is the agent's ability to develop its own algorithms and strategies over time. This not only involves learning from data but also enables the agent to continuously improve itself based on its processing results, learning from past cases. Through iterative self-assessment and refinement, the agent gradually enhances its performance and may even modify its underlying processes to better adapt to changing environments or objectives.

Creativity in agent systems goes beyond mere problem-solving; it includes generating new ideas, methods, or outputs that are both original and valuable. This involves not only developing unique approaches to tackle complex image processing challenges that standard algorithms cannot handle, but also creatively generating content such as artistic transformations, stylizations, or entirely new visual effects.

These are grand visions under higher levels of agency and autonomy. At this stage, exploration of these issues is still quite limited. This paper serves only as an introduction to envisioning these higher levels of intelligence.

