# OpenReview forum: "Position: Agentic Systems Constitute a Key Component of Next-Generation Intelligent Image Processing"
_NeurIPS.cc/2025/Position_Paper_Track — Submitted to NeurIPS 2025 Position Paper Track_

### Official Review · Reviewer_NwSD · 2025-07-28

**Significance:** 2
**Presentation:** 3
**Rating:** 7
**Confidence:** 4

**Summary:**

This paper advocates for adopting agentic system designs as a complementary paradigm to traditional model-centric approaches in image processing. It argues that agentic systems, capable of dynamically selecting and integrating various image processing tools, can enhance generalization, adaptability, and practical application flexibility beyond the capabilities of deep learning models alone.

**Strengths:**

The paper clearly and convincingly argues for the need to shift towards agentic systems in image processing, effectively highlighting current model limitations. It thoroughly discusses the theoretical foundations and practical benefits of agentic systems, backed by relevant literature and detailed conceptual frameworks. The authors also thoughtfully address potential alternative viewpoints.

**Weaknesses:**

The paper could further benefit from concrete empirical examples or preliminary studies demonstrating the practical effectiveness of agentic systems in specific image processing scenarios. Additionally, a discussion on the computational costs and practical feasibility of implementing complex agentic systems in real-world applications would strengthen the argument.

**Questions:**

Can the authors provide concrete examples or preliminary empirical evidence of agentic systems effectively outperforming traditional model-centric approaches in real-world image processing tasks?

How do the authors envision addressing potential computational challenges or efficiency concerns related to deploying agentic systems at scale?

**Alternative Position:**

Yes, and alternative positions are well-considered and addressed by the argument

**Author Identification:**

No.

**Context:**

3

**Discussion:**

3

**Ethics:**

["NO or VERY MINOR ethics concerns only"]

**Position:**

Yes, the paper argues for or against a position related to machine learning.

**Support:**

3

**Thoroughness:**

4

---

### Official Review · Reviewer_RkA2 · 2025-08-09

**Significance:** 2
**Presentation:** 3
**Rating:** 6
**Confidence:** 4

**Summary:**

The paper takes the position that research in the area of image processing should pursue agentic AI as the basis for advanced image processing techniques, as opposed to end-to-end ML-based solutions that try to solve the problem in one pass.  The argument is that an autonomous agent can learn to mix and match different techniques that adapt to the particular input and task.  The authors identify levels of flexibility and autonomy from level 0 (basic tools used by a person) to level 5 (fully autonomous agent able to decide what modifications to make and how to execute them).

**Strengths:**

The paper does a good job of describing a vision for what capabilities and qualities an intelligent image processing system should possess. The taxonomy of image processing systems enables the reader to understand the context and differentiate their vision from existing approaches.

The paper is a well-supported argument for a long-term vision for image processing systems.

When completing the review, one of the most difficult questions to answer was whether this paper was arguing for or against a position in machine learning, or whether it was arguing that a specific technical approach was superior to another.  Given the alternative view discussion and the introduction, it feels like this is arguing for one technical approach over another.  However, these "technical approaches" are very broad, and the argument could be summarized as "in a field that has primarily focused on low-level engineering-driven solutions, a focus on agentic AI could enable revolutionary new capabilities."  So this could be a strength or a weakness.

**Weaknesses:**

1. The alternative position outlined in the paper is both real and yet also feels like a straw man to some degree.  I've marked it as well-considered and addressed, because both the alternative view and the counter-arguments to it are also woven implicitly into the introduction.  But there may be more sophisticated alternative views.

2. While the levels of capability are well-described by the paper. There are no concrete examples given of how those levels would address a specific task.  This makes it challenging for the reader to understand their differences on a specific task.  For example, consider the task of removing a shadow in an image from someone's face.  These would be my interpretations of how the different levels would work

level 0: manual pixel-level manipulation using photoshop like tools
level 1: auto-segmentation or fill capability directed by the user
level 2: auto-pipelining of several steps, quality judged by the user
level 3: iterative exploration with some type of automatic quality evaluation to allow strategy adjustment
level 4: system recalls prior processes that can adapt to new situations
level 5: system predicts the desired modifications and execute them prior to the user seeing the image

**Questions:**

1. It feels like more sophisticated alternative views might exist, including, for example, that a completely autonomous image processing system might be dangerous in the sense that people would never know how the actual original image appeared.  What are the author's thoughts on this?

2. Could the authors provide their vision of a concrete image processing task solved by levels 0 to 5?

**Alternative Position:**

Yes, and alternative positions are well-considered and addressed by the argument

**Author Identification:**

No.

**Context:**

3

**Discussion:**

2

**Ethics:**

["NO or VERY MINOR ethics concerns only"]

**Position:**

Yes, the paper argues for or against a position related to machine learning.

**Support:**

3

**Thoroughness:**

4

---

### Official Review · Reviewer_ffq4 · 2025-08-10

**Significance:** 4
**Presentation:** 3
**Rating:** 5
**Confidence:** 4

**Summary:**

This paper presents a position that the image processing community should focus more on an LLM-based agentic system beyond an end-to-end model.
This is because the agentic AI system can coordinate low-level image processing operators or specialized deep models to plan a better solution like a human.
Besides, this paper also introduces the basic concepts of the AI agent and shows a blueprint of the agentic image processing system, as well as the demanding problems the community should resolve.

**Strengths:**

- Agentic image processing this paper advocated is a promising direction for future research and real-world applications.

- Clear introduction of the background and challenges of previous end-to-end deep image processing models.

- A clear and detailed blueprint of Agentic image processing is given.

- point out several demanding problems, e.g., Cognitive Architecture.

**Weaknesses:**

- The quantitative evidence is lacking here. Hard to know the current status of the LLM-based agentic image system compared to specialized deep models and human performance.

- The agentic system discussed in this paper is too general; better to have specific image processing cases as examples and a focus.

**Questions:**

NA

**Alternative Position:**

Yes, and alternative positions are trivial straw-man arguments

**Author Identification:**

No.

**Context:**

3

**Discussion:**

3

**Ethics:**

["NO or VERY MINOR ethics concerns only"]

**Position:**

Yes, the paper argues for or against a position related to machine learning.

**Support:**

2

**Thoroughness:**

4

---

### Note · Authors · 2025-09-03

**1-11 Submit Again:**

Definitely yes

**1-1 Submission Process:**

2

**1-2 Next Year:**

I would like to see the position paper track introduce a rebuttal phase and provide structured opportunities for discussion between authors and reviewers. This would help clarify misunderstandings and encourage more constructive dialogue. It would also be valuable to offer reviewers more detailed guidelines tailored to position papers, since the evaluation criteria differ from technical contributions. Finally, the review questionnaire could be more carefully designed to capture aspects unique to position papers, such as originality of perspective, clarity of argument, and potential to spark meaningful debate in the community.

**1-4 Interest:**

["Panel discussions with other position paper authors", "Structured debates on controversial topics", "Workshops for developing position papers", "Mentorship programs for early-career researchers"]

**1-5 Thoughtful:**

7

**1-6 Supportive:**

8

**1-7 Technical Aspects Versus Position:**

3

**1-8 Gate Keeping:**

5

**1-9 Camera Ready Changes:**

We will make the following revisions to the camera-ready version in response to reviewer feedback:

- Add both quantitative and qualitative examples to better support our arguments.
- Incorporate a section analyzing safety aspects, as suggested by Reviewer RkA2.
- Include an analysis of computational complexity, following the recommendation of Reviewer NwSD.

**3-1 Review Response1:**

ffq4

**3-2 Reaction To Review1:**

We sincerely thank the reviewers for their comments.

On the claim that our alternative positions are trivial straw-man arguments:

We respectfully disagree. Our central argument is that Agentic Systems will be a core paradigm for future intelligent image processing systems. These systems have not previously received targeted attention because the prevailing view in the community is that “building better and more powerful models will eventually achieve intelligent image processing.” Although rarely stated explicitly, this assumption has guided most research in this area. As we emphasize, the existing paradigm is predominantly model-centric, whereas our contribution highlights the importance and necessity of shifting toward an agentic-system-centric paradigm. This is not a straw-man argument but rather a critical reflection on current practices and their conceptual elevation.

On the lack of quantitative evidence:

We appreciate this concern. There is already a growing body of work that demonstrates the value of agentic systems, and many of these studies contain quantitative evidence supporting our perspective. As a position paper, our primary goal is to articulate the concept, taxonomy, and open problems of agentic systems. Establishing benchmarks and providing detailed quantitative evaluations of specific methods are beyond the scope of this paper. Nevertheless, we emphasize that we are actively working on such quantitative studies and methodological developments, which will be reported in subsequent, more focused publications. In the revised version, we will cite existing literature that provides relevant quantitative evidence.

On including concrete image processing case studies:

We agree that concrete examples can be valuable. Many relevant examples can already be found in existing works. In the camera-ready version, we will incorporate selected representative cases to illustrate and ground the arguments presented in this position paper.

**3-3 Review Response2:**

RkA2

**3-4 Reaction To Review2:**

We thank the reviewers for their thoughtful comments and respond to them in turn.

On whether the paper is supporting or opposing a particular view, or arguing that one technical method is superior to another:

Our position is that Agentic Systems will become a core paradigm for future intelligent image processing. This paradigm has not yet received targeted attention, largely because of a prevailing assumption in the community that “building more powerful and versatile models will eventually achieve intelligent image processing.” Although rarely made explicit, this assumption has guided much of the research to date. As we have argued, the current paradigm is model-centric, whereas we emphasize the importance and necessity of a complementary agentic-system-centric paradigm. Importantly, we are not claiming that this paradigm is “superior” to the model-centric one. Rather, we argue that the two are orthogonal: model-centric approaches have limitations, and agentic systems provide a means to overcome them. Our contribution is to highlight and argue for this underexplored and underemphasized perspective.

On the interpretation of how different levels operate:

We very much appreciate the reviewer’s insight. We will carefully consider this suggestion and incorporate it into our ongoing refinement of the framework.

On the possibility of more complex alternative viewpoints:

We acknowledge that richer perspectives on the overall problem space may emerge, especially given the rapid development of this field. Our position is based on both our understanding of the current state of the research community and our broader vision of where the field is heading. We agree that the safety of fully autonomous image processing systems is an important topic and we are committed to addressing it in the paper.

**3-5 Review Response3:**

NwSD

**3-6 Reaction To Review3:**

We thank the reviewers for their constructive feedback and respond point by point.

On providing concrete examples or preliminary empirical evidence:

We appreciate this suggestion. There is already a growing body of work that demonstrates the value of agentic systems, and many of these studies contain quantitative evidence that supports our perspective. As a position paper, our primary aim is to articulate the concept, taxonomy, and challenges of agentic systems. Designing benchmarks and reporting detailed quantitative evaluations of specific methods are not the main objectives of this paper. Nevertheless, we emphasize that we are actively conducting such quantitative studies and methodological developments, which will be better suited for follow-up papers focusing on particular approaches or datasets. In the revised version, we will include relevant quantitative evidence from existing literature to strengthen the argument.

On addressing potential computational or efficiency challenges in large-scale deployment of agentic systems:

Efficiency is indeed one of our concerns, as it directly impacts cost. However, we believe that at the early stage, efficiency should not be the primary focus. Deployment must first take place in order to create opportunities to reduce costs over time. This development trajectory aligns with many other technological advances, such as deep learning, where concerns about computational challenges were initially valid but were gradually mitigated as the field progressed. We view efficiency improvements in agentic systems in a similar way: the challenges are best addressed through deployment and iterative refinement.

---

### Meta-Review · Area_Chair_ipmz · 2025-09-17

**Rating:** 7
**Confidence:** 5

**Strengths:**

The paper advocates agentic AI systems for image processing, positioning them as a promising and strong alternative to purely end-to-end models. Reviewers all agreed on the clarity of the exposition, especially the blueprint and taxonomy of agentic image processing. The paper presented important open challenges such as cognitive architectures and autonomy levels. Also, the alternative position of this paper (compared with most of the position papers) is strong and offers a good debate ground for the discussion on how/whether agentic AI can further improve.

**Weaknesses:**

While generally not a big issue for a position paper, reviewers are concerned about the lack empirical grounding and the fact that the paper is largely at a conceptual level. More quantitative evidence, case studies, and/or illustrative application scenarios to demonstrate the superiority of agentic systems are suggested. Also, more discussions on the feasibility of agentic AI (computational costs, scalability, and risks) are welcomed.

**Questions:**

Do you think there could be "emergence property" of current agentic AI system in the context of image processing? If yes, please provide some examples.

**Thoroughness:**

5

---

### Decision · Program_Chairs · 2025-09-26

Reject